evolution, genetics, behaviour

genital evolution, sexual conflict, female choice, primary sexual traits, artificial selection, good genes

**Author for correspondence:**
Göran Arnqvist
e-mail: goran.arnqvist@ebc.uu.se

# Direct and indirect effects of male genital elaboration in female seed beetles

Göran Arnqvist[1], Karl Grieshop[1,2], Cosima Hotzy[1], Johanna Rönn[1], Michal Polak[3] and Locke Rowe[2,4]

[1]Animal Ecology, Department of Ecology and Genetics, Evolutionary Biology Centre, Uppsala University, Uppsala, Sweden
[2]Department of Ecology and Evolutionary Biology, University of Toronto, Toronto, Ontario, Canada
[3]Department of Biological Sciences, University of Cincinnati, Cincinnati, OH 45221, USA
[4]Swedish Collegium for Advanced Study, Uppsala University, 752 38 Uppsala, Sweden

GA, 0000-0002-3501-3376; CH, 0000-0003-2145-6157

Our understanding of coevolution between male genitalia and female traits remains incomplete. This is perhaps especially true for genital traits that cause internal injuries in females, such as the spiny genitalia of seed beetles where males with relatively long spines enjoy a high relative fertilization success. We report on a new set of experiments, based on extant selection lines, aimed at assessing the effects of long male spines on females in *Callosobruchus maculatus*. We first draw on an earlier study using microscale laser surgery, and demonstrate that genital spines have a direct negative (sexually antagonistic) effect on female fecundity. We then ask whether artificial selection for long versus short spines resulted in direct or indirect effects on female lifetime offspring production. Reference females mating with males from long-spine lines had higher offspring production, presumably due to an elevated allocation in males to those ejaculate components that are beneficial to females. Remarkably, selection for long male genital spines also resulted in an evolutionary increase in female offspring production as a correlated response. Our findings thus suggest that female traits that affect their response to male spines are both under direct selection to minimize harm but are also under indirect selection (a good genes effect), consistent with the evolution of mating and fertilization biases being affected by several simultaneous processes.

## 1. Introduction

Although our understanding of the evolution of reproductive traits through sexual selection is comprehensive [1,2], some aspects of this process remain unresolved. One such facet is the inner workings of the concerted evolution of seemingly harmful sexual traits and behaviours in the two sexes. Traits that increase fitness in one sex, but decrease it in the other, are termed sexually antagonistic and these traits may be favoured by sexual selection despite causing harm in the other sex [3]. There are an increasing number of examples of such traits in the literature, though there are still relatively few well-worked co-evolutionary examples [3,4]. Under these models, resistance traits in one sex evolve in response to direct selection to reduce the costs incurred by antagonistic traits in the other, which spurs a coevolutionary cycle [5–8]. By contrast, so-called good genes models assume that resistance or preference traits in one sex are shaped by indirect selection, where favoured traits in the opposite sex are assumed to reflect and transmit genetic quality, such that preferred individuals produce highly fit offspring. Despite decades of study, evidence for such good genes effects is mixed [9–15].

These two processes may, however, act in concert [16–19]. This is especially true if exaggeration of the antagonistic trait, resulting from sexual selection, is

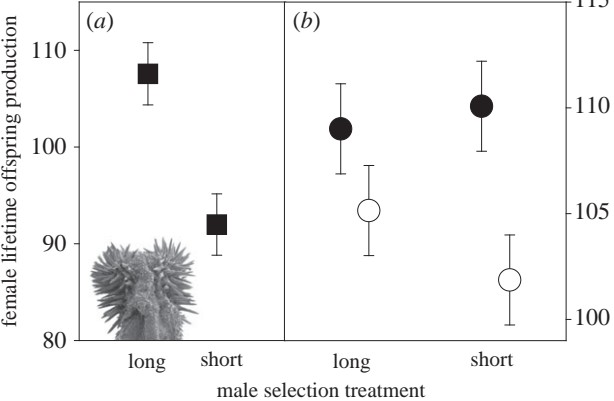

**Figure 1.** Female fitness increases when mated to males from lines selected for long genital spines and females from these same lines have higher fitness, irrespective of their mate. The figure shows effects of selection on male genital spine length on female lifetime offspring production when males from selection lines were mated to (*a*) standard base line females and (*b*) females from the selection lines. As illustrated in (*b*), females from lines selected for long male spines (solid circles) showed higher lifetime offspring production than females from lines selected for short male spines (open circles). Shown are means ± s.e. Insert in the lower-left corner depicts the spine-bearing male genitalia.

in any way costly and therefore comes to reflect the general genetic quality of the bearer. For example, in a polyandrous species, consider the invasion of an antagonistic trait in males that directly harms females during mating interactions, but increases male reproductive success [5]. The evolution of female traits that mitigate these costs may favour further exaggeration of the male trait in a coevolutionary arms race. The evolutionary exaggeration of male traits is predicted to proceed until costs to its bearer offset the benefits of further exaggeration, and given these costs, condition-dependent expression of the trait may evolve [20,21]. With condition dependence, male traits may then reflect the genetic quality of their bearer, which in turn opens up the possibility of good genes effects [18,22,23]. At this stage, females mating to males with exaggerated antagonistic traits may gain indirect benefits through the production of offspring sired by high genetic quality fathers.

One of the most dramatic examples of an antagonistic trait, where both experimental studies and comparative studies of sexually antagonistic coevolution have been undertaken, is the remarkable spiny genitalia of seed beetles (figure 1*a*, inset). These genital spines cause significant internal injuries in the reproductive tract of females [24,25] and in response females have evolved a thickened reproductive tract, a very pronounced immune response and an efficient wound healing to cope with internal injuries [26–28]. Elaboration of the spines can be substantial [26] and it is possible that they have evolved to be costly in males and hence also reflect some aspect of male genetic quality. Although a previous manipulation of juvenile resource quality (old versus new beans) failed to find any significant effects on the expression of genital spines [29], this possibility has some support from observations of reduced genital spine allometry upon release from sexual selection in seed beetles [30]. Further, genital morphology shows at least modest condition-dependent expression in other insects [31–33]. Therefore, it is reasonable to hypothesize that an evolutionary outcome of the sexually antagonistic evolution of these elaborate genitalia may involve a good genes process,

where females gain indirect benefits from having their offspring fathered by males with elaborated genitalia. A recent study of seed beetles demonstrating that a high genetic load for reproductive fitness in males is genetically associated with low female lifetime offspring production [34] supports this possibility. Here, we assess the direct effects of the spines on female fitness and then ask whether exaggerated spines are genetically associated with indirect benefits to females.

Male genital spines, whether manipulated genetically or phenotypically, are favoured through post-mating sexual selection by cryptic female choice in seed beetles [25,35,36], where female traits apparently affect male post-mating fertilization success. However, long spines cause more internal injuries in females and so should bring direct costs to females [24,25,37,38], all else equal. Comparative studies within [25,27] and across [26] species have provided support for such costs and for sexually antagonistic coevolution. It is, however, very difficult to assess the cost of spines to female fitness due to the confounding effects of both female resistance adaptations and male traits that both covary with genital spines [37]. In an effort to overcome these challenges, we first tried to isolate the direct effects of genital spine morphology by revisiting and adding to data from a previously published experiment [35] which used phenotypic engineering (micro-laser ablation) to reduce genital spine length to different degrees in males. These data have the benefit of isolating the effects of phenotypic variation in genital spines, thereby controlling for variation in correlated traits. To estimate the indirect effects of evolutionary elaboration of genital spines, we then present data from new follow-up experiments based on the bidirectional artificial selection lines of male genital spine length previously used by Hotzy *et al.* [35] to study male competitive fertilization success. In these new experiments, males with long and short evolved spines were used to study the effects on the economics of female reproduction.

## 2. Methods

For all experiments reported here, we used the outbred South Indian (SI) stock population of the seed beetle *Callosobruchus maculatus* (Coleoptera, Bruchidae) [39]. Beetles were reared on mung beans (*Vigna radiata*) in climate chambers at 30°C, 55% RH and a 12 : 12 diurnal light cycle at a large population size (*N* > 1000). In order to generate virgin individuals, single beans with larvae were isolated in cell culture well plates prior to the emergence of adult beetles. Experiments were conducted at the University of Cincinnati (work involving spine-ablated males) and the Uppsala University (work involving selection lines). The data on direct costs of spines to females are partly the same data as that reported by [35], but the data and analyses have here been supplemented with new measures of male and female body size (see below).

The experiments on the effects of the evolution of male spines in females reported here represent distinct, novel and entirely independent experiments that employ the artificial selection lines previously used to assess a distinct question (i.e. the role of male genital spine length for male competitive reproductive success [35]). Hence, although the selection lines used are the same, the experiments described here are an entirely distinct set of experiments from those reported previously [35]. In other words, the current follow-up experiments were conducted after those reported in [35], with the present aim being to elucidate the role of male genital spine length on female reproductive

fitness. In addition, we present a novel effort at characterizing the genetic architecture of male genital spines.

## (a) Direct costs of spines

Hotzy *et al.* [35] used a micro-laser ablation system to phenotypically manipulate the ventral genital spines. Briefly, they created two treatment groups, one which had 30 ventral spines of the aedegus shortened (S; relatively short spines) and a second which only had 10 ventral spines shortened (L; relatively long spines). Although they found that males themselves were not differentially affected by the surgical laser treatment *per se*, in terms of post-treatment water/food consumption or lifespan, they showed that males with relatively long spines enjoyed a higher competitive fertilization success [35].

Hotzy *et al.* [35] also estimated the direct effects of spines in females. Briefly, standard stock females, 24–48 h post-eclosion, were first mated to a standard male (day 1) and then to a second and focal male from the L or S group (day 3) ($N_L = 24$; $N_S = 24$). Females were allowed to oviposit on mung beans both between the first and the second mating and then for 7 days after the second and focal mating (at which time females were dead or post-reproductive). Female fecundity was quantified by counting the number of eggs deposited on beans. Hotzy *et al.* [35] found that female fecundity after the second mating was somewhat lower in females mated to L males compared to S males, but not significantly so ($p = 0.091$). However, this analysis did not take the effects of male and female body size into account.

Here, we revisited the individuals used in the experiment of Hotzy *et al.* [35]. We measured the body size (elytra length) of all focal individuals used (females and their second mates) with a digital calliper under a dissecting microscope, in an effort to improve the previous analysis by enabling us to control statistically for effects of male and female body size on female fecundity. We predict a negative direct effect of male spine length on female fecundity, and tested this in an analysis of covariance, using the number of eggs laid after the second mating as our response variable. Male treatment was our focal factorial variable and the number of eggs laid between matings, female body size and focal male body size were considered nuisance variables and used as covariates.

## (b) Effects of the evolution of male spines

We employed the same selection lines that were previously used to assess the role of male genital spine length for male competitive reproductive success [35], and refer to ref. [35] for details on materials and procedures used to generate these lines. Briefly, replicated selection lines (population size: $N = 100$ beetles per line) were selected for long (L; $N = 3$ lines) or short (S; $N = 3$ lines) genital spines during five consecutive generations. In each generation, we selected those 33% of the males with the longest/shortest spines to propagate the next generation. Three pairs of replicate lines were set up in three consecutive cohorts of the baseline population, which were kept in temporal succession (i.e. three time-staggered cohorts, two lines in each). The cohort was thus used as a factorial variable in our statistical analyses. At generation six, male ventral spines in the L selection lines were some 14% longer than those in the S lines.

Using the above six selection lines, we set up two experiments, additional to the original ones reported in [35]. Both the original and additional experiments were run in 2012. We first asked how L and S males affected female lifetime offspring production in females from the base line (i.e. the unselected population used to found the S and L selection lines). Lines were reared for two generations under common garden conditions (i.e. no selection) prior to these assays. We then set up $N = 20$ replicate assays per selection line (total $N = 20 \times 6 = 120$). In each such replicate assay, a randomly selected virgin base line female was introduced with two virgin selection line males (all 24–48 h post-eclosion) in a Petri

dish (6 cm Ø) provided with 8 g of mung beans. Dishes were then kept under rearing conditions until parental individuals were dead and all offspring had emerged, at which point the number of offspring produced was recorded. We tested for the effects of our spine selection treatment by a paired *t*-test of mean female offspring production per selection line (mean of the $N = 20$ assays per line), comparing the S and L lines within each cohort ($N = 3$ cohorts; $N = 2 \times 3 = 6$ lines).

We then assessed the independent effects of male and female selection history on female fitness, by pairing L and S individuals in an orthogonally crossed fashion within each of the three cohorts. Hence, male × female combinations were L × L, L × S, S × L and S × S within each cohort and we set up eight replicates of each of the four crosses in each cohort. Hence, the total number of replicates was $N = 96$ (i.e. $4 \times 8 \times 3$). In each replicate, five virgin males and five virgin females (all 24 h post-eclosion) were introduced together in a Petri dish (9 cm Ø) containing 35 g of mung beans. Dishes were kept under rearing conditions until parental individuals were dead, at which point they were removed and their body size (elytra length) measured. Dishes were subsequently stored under rearing conditions until all offspring had emerged, at which point the number of offspring produced was recorded. These data were analysed, using REML estimation, in a linear mixed model of mean female egg production (averaged over the eight replicates in each cell), where cohort was a random-effects factor. The selection treatment origin of females and males (S or L), as well as their interaction, were fixed effects factors. In addition, to account for the potential effects of body size, we also regressed mean female offspring production on mean female body size per replicate and then used mean residual offspring production per cell in our design as our response variable in an analogous mixed model. Again, the cohort was a random-effects factor and the selection treatment origin of females and males (S or L) were fixed effects factors.

We note that hatching rate and egg–adult survivorship is generally high in *C. maculatus* and the conditions under which these assays were run involved very low levels of larval competition, such that egg–adult survivorship would have been very high (>90%) [40]. This is also evident from the fact that mean per female lifetime offspring production in our assays was 103.9 (s.e. = 1.43) offspring, which is very closely aligned with previous data on the maximal per capita average lifetime female egg production in this population (mean 101.2; s.e. = 5.3 [41]). Hence, variance in female fitness in our assays should primarily reflect variation in female fecundity, rather than variation in the juvenile survival of their offspring.

## (c) Genetic architecture of male genital spines

In a supplementary effort to (i) validate the presence of additive genetic variation in male genital spines [35], (ii) to determine whether spine phenotypes are condition dependent and (iii) to assess the genetic covariance between spine length and overall body size, we performed a new quantitative genetic breeding experiment. Here, replicated sets of full-sib offspring from the base line population were reared on either of four different food types (lentils, chick peas, adzuki beans and mung beans) differing in nutritional quality and were phenotyped for genital spine length, mid-leg claw length and body size upon adult emergence. We refer to the electronic supplementary material for a detailed description of the methods and analyses of this non-focal experiment.

# 3. Results

## (a) Direct costs of spines to females

Following a log transformation of female fecundity to normalize residuals and stabilize variance, one deviant female

royalsocietypublishing.org/journal/rspb　　Proc. R. Soc. B **288**: 20211068

showed a standardized residual fecundity of $R = -5.6$ and was deemed an outlier and hence removed prior to fitting our inferential model. This female laid only a single egg after her second mating, which was well outside the range of all other females (range = 13–87; average = 42.2, s.d. = 15.6).

As predicted, we found that females mated with males with relatively long spines produced on average some 17% fewer eggs after their second mating (LS mean = 38.3, s.e. = 2.51) than did those mated to males with shorter spines (LS mean = 46.3, s.e. = 2.56). An analysis of covariance showed a significant effect of genital spine treatment on female egg production ($F_{1,42} = 5.09$, $p = 0.029$). The homogeneity of slopes assumption of this model was verified as the three interactions between the covariates (female body size, male body size and eggs laid between matings) and the spine length treatment did not collectively improve model fit to data ($F_{3,39} = 0.92$, $p = 0.44$), and interactions were thus not included in the inferential model. The three covariates collectively improved model fit ($F_{3,42} = 13.32$, $p < 0.0001$). In particular, the effect of female body size on female fecundity was sizeable ($F_{1,42} = 12.1$, $p = 0.001$). The residuals of our inferential model were well behaved (Levene's test for homogeneity of variance; $p = 0.534$. Kolmogorov–Smirnov test for normality; $p = 0.326$). This result provides evidence for a direct cost to females of male genital spines *per se*.

## (b) Direct effects on female fitness of the evolution of male spines

We have previously shown that mating rate does not differ significantly between base line females paired with L and S selection line males [35]. The new experiments reported here showed that average lifetime offspring production, primarily reflecting female fecundity (see above), was some 17% higher in base line females mated to L males than in those mated to S males (figure 1a) (paired *t*-test; $t_2 = 7.72$, $p = 0.016$) when kept for life with males. A previous experiment performed in twice-only mated females, using males from the same lines, found a non-significant effect in the same direction [35]. We note that male body size was about 0.8% smaller in the L lines compared to the S lines (mean elytra length; S: 2.04, s.e. = 0.011 mm, L: 2.03, s.e. = 0.007 mm) [35], consistent with the negative genetic correlation found between male spine length and female body size in this population (see below).

In our second assay, we instead mated males and females from our selection lines in a fully orthogonally crossed manner, and assayed female lifetime offspring production. Again, females mated with L males enjoyed slightly higher lifetime offspring production (figure 1b) (L: 107.1, s.e. = 1.53; S: 105.9, s.e. = 2.24), but not significantly so in this experiment (table 1; effect of male selection line origin).

## (c) Indirect effects on female fitness of the evolution of male spines

The second assay showed that selection on male spine length significantly affected female fitness as a correlated response: females derived from L lines showed a 6% elevation in lifetime offspring production compared to females from S lines (figure 1b) (table 1; effect of female SLO), averaged over all mates, thus demonstrating a positive genetic covariance between male genital morphology and female lifetime offspring production. Female body size was positively related to

**Table 1.** Tests for fixed effects of selection line origin (SLO; S or L) of males and females on female lifetime offspring production.

| source | ndf | ddf | F-value | p-value |
|---|---|---|---|---|
| male SLO | 1 | 6 | 0.30 | 0.604 |
| female SLO | 1 | 6 | 8.92 | 0.024 |
| male SLO × female SLO | 1 | 6 | 1.16 | 0.323 |

lifetime offspring production ($R^2 = 0.11$, $F_{1,94} = 12.41$, $p < 0.001$) and females in L lines evolved to be 1.1% larger than females in S lines (mean elytra length; S: 2.10, s.e. = 0.005 mm, L: 2.12, s.e. = 0.004 mm) [35], consistent with the positive intersexual genetic correlation found between male spine length and female body size (see below). Therefore, the indirect effect in females of the male spine selection treatment could potentially be the result of an evolutionary increase in female body size. However, statistically removing the effects of female body size on offspring production yielded very similar results (male SLO: $F_{1,7} = 0.14$, $p = 0.717$; female SLO: $F_{1,7} = 6.511$, $p = 0.038$). This assay thus demonstrated that there is a positive genetic association between male genital spine length and female size-specific fitness.

## (d) Genetic architecture of male genital spines

Estimates of broad and narrow sense heritability of genital spine length ranged between $h^2 = 0.10$–$0.54$ and $h^2 = -0.07$ to 0.51, respectively, across the four food treatments, confirming the presence of additive genetic variation but also suggesting that the genetic architecture of spines differ across environments. The expression of genital spines was also significantly dependent upon food quality ($F_{3,21.6} = 3.92$, $p = 0.022$), but less so compared to body size ($F_{3,23.2} = 54.49$, $p < 0.001$). Thus, in this sense, male body size was more condition dependent than male genital spine length. Finally, we assessed sex-specific genetic covariance between body size and genital spine length by relating phenotypic variation in the two traits across generations. These models revealed a negative covariance in males ($F_{1,34} = 11.61$, $p = 0.002$) and a positive covariance in females ($F_{1,34} = 12.27$, $p = 0.001$). We refer to the electronic supplementary material for a detailed description of the results of this non-focal experiment.

## 4. Discussion

Our experiments on a well-known model system of sexually antagonistic coevolution reveal a surprisingly complex economy of costs and benefits to females of injurious male genitalia. These include direct costs but also direct benefits and, surprisingly, indirect benefits associated with the evolutionary elaboration of genitalia in males. This complexity suggests that at least three distinct processes of sexual selection, which are sometimes viewed as incompatible, are all operating in a single system [29]. Specifically, whatever the female traits may be that lead to a reproductive advantage for long-spined males [25,35], these female traits are apparently shaped by two opposing sources of direct selection as well as indirect selection.

First, our phenotypic engineering experiment confirmed that shortening of these male-benefit spines has a direct positive effect on female fitness, previously only inferred from comparative studies [26,27]. Previous work has demonstrated that the spines cause injuries to the female reproductive tract [24,25,37] and induce an immune response in females [27,28,42]. Genital spine length in males is correlated with female reproductive tract morphology across populations [27] and species [26], where an evolutionary thickening of the reproductive tract is associated with longer spines. The evolution of spines is, however, associated with a number of correlated traits and isolating their effect requires experimental manipulation. This is likely especially true as females should be well adapted to resist genital spines and any effects are thus likely to be small [3,37]. Here, we employed phenotypic engineering of spine length to extend the natural range of spine length variation and demonstrate that genital spines *per se* do indeed impose a cost on females implying that there is direct selection on females against mating with long-spined males, all else equal.

However, our experiments demonstrate that all else is apparently *not* equal. Male–female interactions carry multiple interacting costs and benefits to females in this model system [43], several of which are mediated through seminal fluid substances provided by males [44–46]. The artificial selection experiment suggested that the evolutionary exaggeration of genital spines results in the correlated evolution of traits in males that confer a direct fecundity advantage to females who mate with males with longer spines. Moreover, this direct net benefit apparently occurred despite the fact that the spines *per se* tend to reduce female fecundity (see above).

There are six reasons to believe that this positive direct effect was mediated primarily through the seminal fluid transferred to females. First, males transfer a very large (5–8% of male bodyweight) and costly [47,48] ejaculate to females in this species. Second, although ejaculate size did not differ significantly between S and L males [35], the *C. maculatus* ejaculate is now known to contain greater than 300 different proteins [49,50] several of which demonstrably affect female fecundity [44–46], and S and L males may have differed in ejaculate composition. Third, a previous proteomic study demonstrated an association between ejaculate composition and components of female fecundity in this species [44]. Fourth, male spine length is known to affect the uptake of seminal fluid substances into the female body cavity after mating [35]. Fifth, ejaculate size and composition is known to exhibit genetic variation [44,51,52]. Sixth, a shift in allocation towards higher ejaculate quality in L males is consistent with our observation that a correlated response to artificial selection for increased spine length was, if anything, a decrease in male body size, which is also reflected in a negative genetic covariance between male body size and genital spine length. The phenotypic correlation between genital spine length and male body size tends to be weakly positive both within ([29]; $r = 0.17$) and between ([25]; $r = 0.21$) populations of this species. Trade-offs between investment in primary sexual traits, ejaculates and non-sexual traits have been observed both across *Callosobruchus* species [53–55] and in several other insect species [56]. Although we show that genital spine length is condition dependent in *C. maculatus*, others have found no effect of food resource quality on spine length using a different environmental treatment [29]. This suggests that the degree of condition dependence may vary across populations and may, additionally, differ depending on the type of environmental challenge.

Finally, and perhaps most surprisingly, we found that selection for increased genital spine length in males resulted in an increase in female lifetime offspring production as a correlated response. These data are thus a rare direct demonstration of a positive intersexual genetic correlation between a sexually selected trait in males and female fitness, a key prerequisite for good genes effects. In terms of the underlying genetic mechanism responsible for this correlation, we see at least two non-mutually exclusive possibilities. One is that these sexually selected traits in males are costly and thus a reflection of male quality which in turn is reflected in the elevated fitness of their female descendants. That exaggerated sexually selected traits come to reflect the genetic quality of their bearer, through 'genic capture', is an essential assumption of the good genes process because it aligns male reproductive fitness with non-sexual fitness components [2,23]. This possibility is supported by the effects of induced deleterious mutations on reproductive fitness being to some extent shared between the sexes in seed beetles [57] and by the fact that the naturally occurring mutation load on fitness in male seed beetles is indeed negatively genetically associated with female fitness [34]. We note, however, that the strength of condition-dependent expression of genital spines is apparently variable (see above) and the relative importance of good gene effects is thus likely to vary across populations and environments in this species, a prediction underscored by the fact that the genetic correlation between male and female reproductive fitness is known to vary from positive to negative [58–61]. The accumulated insights into male–female coevolution in this relatively well-studied model system have so far, thus, revealed a remarkable complexity. A second possibility is that selection for long genital spines resulted in the evolution of increased allocation to reproduction in both sexes, reflected in the elaboration of spines and ejaculate substances in males and (as a correlated response) in elevated fecundity in females. This shared life-history trade-off scenario is supported by the positive direct effects of L males on female fecundity and by the negative genetic correlation between body size and spine length in males, but less so by the positive genetic correlation between female body size and male spine length (electronic supplementary material). It is also consistent with the fact that primary sexual traits in males and females show correlated evolution across *Callosobruchus* species [62] and comparative data suggest trade-offs between investment in primary sexual traits and other competing life-history demands, such as immunity [28], in this genus [55]. Irrespective of the underlying genetic mechanism, however, our results show that genetic variation associated with spine length in males is positively related to genetic variation in female lifetime offspring production, under the conditions in which our assays were performed.

The results presented here represent a rare case supporting one of the prerequisites of the good genes process: a positive genetic covariance between a male trait under (cryptic) female choice and female fitness or viability (e.g. [63]). What makes our study especially unusual is that the sexually selected trait, elongated genital spines, also has sexually antagonistic effects. Although sexually antagonistic coevolution and good genes are often considered as incompatible, our results shed new light on the complexity of male–female coevolution: there is the potential for the good genes process to act whenever sexual selection leads to the exaggeration of favoured traits [16–18]. We suggest that our results provide an illustration of the fact that we should expect various forms of selection on (cryptic) female choice to act

simultaneously. The challenge now, perhaps, lies in unravelling their relative importance [18,64].

Data accessibility. All data reported in this paper have been published in Mendeley Data and are available at http://dx.doi.org/10.17632/tp7mhmtyxy.1.

Authors' contributions. G.A.: conceptualization, formal analysis, funding acquisition, methodology, project administration, supervision, visualization, writing—original draft, writing—review and editing; K.G.: data curation, formal analysis, investigation, writing-review and editing; C.H.: conceptualization, data curation, investigation, methodology, writing—review and editing; J.R.: conceptualization, data curation, investigation; M.P.: conceptualization, investigation, methodology, resources, supervision, writing—review and editing; L.R.: conceptualization, funding acquisition, writing—original draft, writing—review and editing.

All authors gave final approval for publication and agreed to be held accountable for the work performed therein.

Competing interests. The authors declare that they have no competing interests.

Funding. This study was supported by grants from the European Research Council (GENCON AdG-294333), the Swedish Research Council (grant nos. 2014-4523; 2019-03611) and FORMAS (grant no. 2018-00705) to G.A. L.R. was supported by the Natural Sciences and Engineering Research Council of Canada and the Swedish Collegium for Advanced Study. M.P. was supported by National Science Foundation (NSF) USA grant no. DEB-1654417. K.G. was supported by the Swedish Research Council (grant no. 2018-06775).

Acknowledgements. We thank all members of the GENCON lab group, Uppsala University, for helpful discussions.

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
