## [Peer Review File · Proceedings of the Royal Society B: Biological Sciences]

Review History

RSPB-2021-0578.R0 (Original submission)

Review form: Reviewer 1

Recommendation

Major revision is needed (please make suggestions in comments)

Scientific importance: Is the manuscript an original and important contribution to its field?

Excellent

General interest: Is the paper of sufficient general interest?

Good

Quality of the paper: Is the overall quality of the paper suitable?

Good

Is the length of the paper justified?

Yes

Should the paper be seen by a specialist statistical reviewer?

No

Do you have any concerns about statistical analyses in this paper? If so, please specify them explicitly in your report.

Yes

It is a condition of publication that authors make their supporting data, code and materials available - either as supplementary material or hosted in an external repository. Please rate, if applicable, the supporting data on the following criteria.

Is it accessible?

Yes

Is it clear?

Yes

Is it adequate?

Yes

Do you have any ethical concerns with this paper?

No

Comments to the Author

The manuscript reports a series of experiments examining the direct and indirect effects that male genital spines have on female fitness in the seed beetle *Callosobruchus maculatus*. The conflict between males and females that arises due to this harmful genital trait have been well-studied in this species, though results are mixed. By experimentally shortening male spines, the authors show for the first time that males with long spines significantly reduce female fecundity. However, artificial selection on male spine length finds the opposite result- reference females mating to males with long spines have higher fecundity, and selection for increased male spine length led to a correlated increase in female fecundity. These results suggest that females actually gain indirect benefits to mating with males with long spines, and that sexual coevolution in this species is driven by both sexual antagonism and good genes sexual selection.

Overall, this is a very well-presented study describing an elegant series of experiments testing these ideas. Further, the results significantly improve our understanding of sexual coevolution in this model species, and will be relevant to the study of sexual conflict and sexual selection more generally. However, I have some major concerns with how the study is presented, and how it links to previous studies, which I feel the authors need to address.

Main comments

1. This study builds heavily from a study from this group from 2012 (Hotzy et al., 2012). This means that several of the approaches used in this study are not novel (though some of the results are). Specifically, the previous study used the same phenotypic engineering of male spine length (but found no significant difference between manipulated and control males), and compared the same experimental evolution lines (though female fitness was not recorded in the earlier study). This in and of itself is not a problem. However, for me this relationship was not made very transparent in the present paper, at least in the introduction (e.g. lines 105-111). I would go so far as to say that, for a reader not familiar with this previous study, this link would very much not be apparent from a casual read of the paper. So, to give due credit to this earlier work, and to put this study in its proper context, I think the links between these two studies needs to be much clearer in the introduction, and throughout.
2. Related to above, at several points that manuscript mentions using partly the same data from Hotzy et al., 2012 (lines 128-130, 240-242). However, it is not clear to me exactly which data were re-used. This needs to be made clearer here and in the introduction, along with a justification for why this way done.
3. More generally, I found the discussion of previous empirical studies in this species to be

lacking. There is now a large literature focusing on sexual conflict in this species, and the results are complex. However, I feel some discussion of this complexity is useful for putting these results into context. For example, Cayetano & Bonduriansky (2015) explicitly test whether male genital spine length was condition dependent, but did not find an effect. This result is not mentioned when considering condition-dependence in the introduction (lines 88-89), or in the discussion (line 323). I would therefore like to see an expanded discussion of this and other relevant studies (incl. Hotzy et al., 2012) to better put these results into context. Including this context would not diminish the results of this study- indeed, the surprising result of this study (that both sexual conflict and good-genes effects may influence spine evolution) may partly explain why results of previous studies have been so mixed.

4. I'm not entirely comfortable with the removal of the two outlier females in the phenotypic engineering experiment (lines 162-164), especially given that the result this pertains to is barely significant. Is $R > 2.5$ a common metric? Regardless, I would prefer you to present the results without removing these females

5. Several interesting results are presented in the supplementary material, but it is not obvious why. Does the manuscript as it stands overrun the Proc B space limits? If not, I think you should add these methods and results to the main text

Other comments

1. Line 104: remove extra 'of'

2. Lines 181-183: why were two males used here?

3. Lines 169-171: When were the experiments using the sexual-selection lines done? Around 2012?

4. Line 213: please list the maximal value so readers do not have to access the reference

5. Lines 238-239: why no figure for this significant effect?

6. Lines 290-292: other authors have suggested that negative effects on female fitness may be hard to detect because they are very small when the species is at a coevolutionary equilibrium. Do these results counter that argument?

Review form: Reviewer 2

Recommendation

Major revision is needed (please make suggestions in comments)

Scientific importance: Is the manuscript an original and important contribution to its field?

Good

General interest: Is the paper of sufficient general interest?

Good

Quality of the paper: Is the overall quality of the paper suitable?

Marginal

Is the length of the paper justified?

No

Should the paper be seen by a specialist statistical reviewer?

Yes

Do you have any concerns about statistical analyses in this paper? If so, please specify them explicitly in your report.

No

It is a condition of publication that authors make their supporting data, code and materials available - either as supplementary material or hosted in an external repository. Please rate, if applicable, the supporting data on the following criteria.

Is it accessible?

N/A

Is it clear?

N/A

Is it adequate?

N/A

Do you have any ethical concerns with this paper?

No

Comments to the Author

RSPB 2021-0578

Direct and indirect effects of male genital elaboration in seed beetles

This paper examines the effect of male genital spine morphology on direct and indirect female fitness. The results are potentially very important because the effects of sexual conflict have been wrongly assumed to be only through direct fitness effects, while clearly there can be an impact on indirect fitness. The paper relies on spine ablation data from a previous published experiment adding body size as a novel aspect of the work, and from selection experiments for short and long spines. While the questions asked in this manuscript are indeed exciting, I had a hard time following the different experiments and their results. There is a lot of simplification here which seems necessary to make the paper more readable, but this comes at the expense of clarity of which crosses the authors are talking about in specific areas. I had to re-read the methods and result several times to try and figure out exactly what was done here.

In addition, I have a hard time with the presentation of both direct costs and direct benefits, in that the direct benefits results come from experiments where females were kept with the same male in monogamy for life, a situation that is highly unlikely in nature, and may have resulted in males modifying other behaviors or seminal fluids, that would mitigate the damage that is normally inflicted by the spines in a single mating, and the authors present good evidence for this in the discussion. That means that the result of increased in offspring production could simply be the result of male modification of ejaculate fluid and have nothing to do with spine length. The section on indirect fitness is relies on female offspring production which is indistinguishable from fecundity as measured for direct benefits, with no justification of how these are interpreted as being different. While I fully understand the limitations of measuring the fitness of offspring produced by the experiments, that would have been the appropriate measure, and in that sense the results are oversold.

The full results of all the statistical models should be included in supplementary materials.

Line 105-106: How does this experiment overcome these challenges though? You are still only manipulating males so the confounding effects of female resistance and other male behaviors remain.

Lines 127-128. If the experiments are distinct, how can the data be the same? This is very confusing!

Line 139-140. This study should stand on its own. Please provide brief description of those procedures using the reference for further details, not instead of an explanation.

Line 163. Please conduct the analyses with and without your outliers. Outliers are part of the normal biological variation and in an experiment with relatively small sample sizes such as this

one, they may change the outcome of the statistics altogether.

Lines 180-188. What is baseline female? A female from a non-selected line? And then what type of selected males? S, L both? There are 20 replicates per selection and then you selected one female to mate with males from each of these replicates? Why are there six lines? I thought there were 2 lines with 20 replicates each? Confusing! Please describe better.

Line 217. The questions that were meant to be addressed by this experiment are all important and relevant to the interpretation of the current results presented in this paper. This experiment should not be part of the supplementary materials, but should be included in full in the manuscript.

Line 233. Name the covariates here. I believe they are male body size, female body size and number of eggs laid between matings?

Line 248: Average lifetime fitness production is 17% higher in L mated females than S mated females? How does that happen when the females produce 15% fewer eggs when mated with L males?

Line 263: Females from L lines had a 6% higher production of offspring regardless of the phenotype of the male with whom they mated right?

Line 269-270. How would this genetic correlation come to be? It could be that in lines where males have longer spines, female defenses such as kicking become more important and larger females are better at kicking...

The "direct benefits" need further justification. How likely would it be in nature that a female would become fully monogamous and end up mating with a single male per life over multiple reproductive bouts? This is one of the confounding factors that the authors mentioned at the beginning

Line 286-287: What are those female traits? Important to include here.

Line 290: the phenotypic engineering did not elongate the spines! It shortened them!! Am I missing something here?

Line 302-327: This explanation confirms my suspicion that males are changing the composition of their ejaculate when they are in this artificially enforced monogamy for a lifetime situation, and therefore the effect of the spine length is completely unknown! The increase in offspring production by females mated to Long spined males for life could be mediated by changes in ejaculate fluid as argued here.

Line 334-336. Exactly the problem here. The indirect benefit has to be through the descendants, and that was not shown in this experiment. You measured the same thing as direct and indirect benefits.

Line 368-371: While I agree wholeheartedly with the idea, this is not shown in this study.

Decision letter (RSPB-2021-0578.R0)

18-Apr-2021

Dear Professor Arnqvist:

I am writing to inform you that your manuscript RSPB-2021-0578 entitled "Direct and indirect effects of male genital elaboration in female seed beetles" has, in its current form, been rejected for publication in Proceedings B.

This action has been taken on the advice of referees, who have recommended that substantial revisions are necessary. With this in mind we would be happy to consider a resubmission, provided the comments of the referees are fully addressed. However please note that this is not a provisional acceptance.

Sincerely,
 Professor Gary Carvalho
 mailto: proceedingsb@royalsociety.org

Associate Editor
 Board Member: 1
 Comments to Author:

The two reviewers were somewhat mixed in their enthusiasm, though both raise concerns over how much of the data and analyses presented here are novel, and how much has been previously published. I was also somewhat confused by this, and without a clearer understanding, I think it is quite difficult to determine the novelty presented in this manuscript by the authors. Both reviewers also ask for a more thorough context of previous empirical findings. Reviewer 1 also raises concerns about the removal of two outlier females from the analysis. Reviewer 2 is far more critical, and raises key points about the limits that the experimental design place on the interpretation. These are, in my opinion, valid, especially the lack of distinction between female fecundity and female offspring production.

Reviewer(s)' Comments to Author:
 Referee: 1

Comments to the Author(s)

The manuscript reports a series of experiments examining the direct and indirect effects that male genital spines have on female fitness in the seed beetle *Callosobruchus maculatus*. The conflict between males and females that arises due to this harmful genital trait have been well-studied in

this species, though results are mixed. By experimentally shortening male spines, the authors show for the first time that males with long spines significantly reduce female fecundity. However, artificial selection on male spine length finds the opposite result- reference females mating to males with long spines have higher fecundity, and selection for increased male spine length led to a correlated increase in female fecundity. These results suggest that females actually gain indirect benefits to mating with males with long spines, and that sexual coevolution in this species is driven by both sexual antagonism and good genes sexual selection.

Overall, this is a very well-presented study describing an elegant series of experiments testing these ideas. Further, the results significantly improve our understanding of sexual coevolution in this model species, and will be relevant to the study of sexual conflict and sexual selection more generally. However, I have some major concerns with how the study is presented, and how it links to previous studies, which I feel the authors need to address.

Main comments

1. This study builds heavily from a study from this group from 2012 (Hotzy et al., 2012). This means that several of the approaches used in this study are not novel (though some of the results are). Specifically, the previous study used the same phenotypic engineering of male spine length (but found no significant difference between manipulated and control males), and compared the same experimental evolution lines (though female fitness was not recorded in the earlier study). This in and of itself is not a problem. However, for me this relationship was not made very transparent in the present paper, at least in the introduction (e.g. lines 105-111). I would go so far as to say that, for a reader not familiar with this previous study, this link would very much not be apparent from a casual read of the paper. So, to give due credit to this earlier work, and to put this study in its proper context, I think the links between these two studies needs to be much clearer in the introduction, and throughout.
2. Related to above, at several points that manuscript mentions using partly the same data from Hotzy et al., 2012 (lines 128-130, 240-242). However, it is not clear to me exactly which data were re-used. This needs to be made clearer here and in the introduction, along with a justification for why this way done.
3. More generally, I found the discussion of previous empirical studies in this species to be lacking. There is now a large literature focusing on sexual conflict in this species, and the results are complex. However, I feel some discussion of this complexity is useful for putting these results into context. For example, Cayetano & Bonduriansky (2015) explicitly test whether male genital spine length was condition dependent, but did not find an effect. This result is not mentioned when considering condition-dependence in the introduction (lines 88-89), or in the discussion (line 323). I would therefore like to see an expanded discussed of this and other relevant studies (incl. Hotzy et al., 2012) to better put these results into context. Including this context would not diminish the results of this study- indeed, the surprising result of this study (that both sexual conflict and good-genes effects may influence spine evolution) may partly explain why results of previous studies have been so mixed.
4. I'm not entirely comfortable with the removal of the two outlier females in the phenotypic engineering experiment (lines 162-164), especially given that the result this pertains to is barely significant. Is $R > 2.5$ a common metric? Regardless, I would prefer you to present the results without removing these females
5. Several interesting results are presented in the supplementary material, but it is not obvious why. Does the manuscript as it stands overrun the Proc B space limits? If not, I think you should add these methods and results to the main text

Other comments

1. Line 104: remove extra 'of'
2. Lines 181-183: why were two males used here?
3. Lines 169-171: When were the experiments using the sexual-selection lines done? Around 2012?
4. Line 213: please list the maximal value so readers do not have to access the reference
5. Lines 238-239: why no figure for this significant effect?

6. Lines 290-292: other authors have suggested that negative effects on female fitness may be hard to detect because they are very small when the species is at a coevolutionary equilibrium. Do these results counter that argument?

Referee: 2

Comments to the Author(s)

RSPB 2021-0578

Direct and indirect effects of male genital elaboration in seed beetles

This paper examines the effect of male genital spine morphology on direct and indirect female fitness. The results are potentially very important because the effects of sexual conflict have been wrongly assumed to be only through direct fitness effects, while clearly there can be an impact on indirect fitness. The paper relies on spine ablation data from a previous published experiment adding body size as a novel aspect of the work, and from selection experiments for short and long spines. While the questions asked in this manuscript are indeed exciting, I had a hard time following the different experiments and their results. There is a lot of simplification here which seems necessary to make the paper more readable, but this comes at the expense of clarity of which crosses the authors are talking about in specific areas. I had to re-read the methods and result several times to try and figure out exactly what was done here.

In addition, I have a hard time with the presentation of both direct costs and direct benefits, in that the direct benefits results come from experiments where females were kept with the same male in monogamy for life, a situation that is highly unlikely in nature, and may have resulted in males modifying other behaviors or seminal fluids, that would mitigate the damage that is normally inflicted by the spines in a single mating, and the authors present good evidence for this in the discussion. That means that the result of increased in offspring production could simply be the result of male modification of ejaculate fluid and have nothing to do with spine length.

The section on indirect fitness is relies on female offspring production which is indistinguishable from fecundity as measured for direct benefits, with no justification of how these are interpreted as being different. While I fully understand the limitations of measuring the fitness of offspring produced by the experiments, that would have been the appropriate measure, and in that sense the results are oversold.

The full results of all the statistical models should be included in supplementary materials.

Line 105-106: How does this experiment overcome these challenges though? You are still only manipulating males so the confounding effects of female resistance and other male behaviors remain.

Lines 127-128. If the experiments are distinct, how can the data be the same? This is very confusing!

Line 139-140. This study should stand on its own. Please provide brief description of those procedures using the reference for further details, not instead of an explanation.

Line 163. Please conduct the analyses with and without your outliers. Outliers are part of the normal biological variation and in an experiment with relatively small sample sizes such as this one, they may change the outcome of the statistics altogether.

Lines 180-188. What is baseline female? A female from a non-selected line? And then what type of selected males? S, L both? There are 20 replicates per selection and then you selected one female to mate with males from each of these replicates? Why are there six lines? I thought there were 2 lines with 20 replicates each? Confusing! Please describe better.

Line 217. The questions that were meant to be addressed by this experiment are all important and relevant to the interpretation of the current results presented in this paper. This experiment should not be part of the supplementary materials, but should be included in full in the manuscript.

Line 233. Name the covariates here. I believe they are male body size, female body size and number of eggs laid between matings?

Line 248: Average lifetime fitness production is 17% higher in L mated females than S mated females? How does that happen when the females produce 15% fewer eggs when mated with L males?

Line 263: Females from L lines had a 6% higher production of offspring regardless of the phenotype of the male with whom they mated right?

Line 269-270. How would this genetic correlation come to be? It could be that in lines where males have longer spines, female defenses such as kicking become more important and larger females are better at kicking...

The "direct benefits" need further justification. How likely would it be in nature that a female would become fully monogamous and end up mating with a single male per life over multiple reproductive bouts? This is one of the confounding factors that the authors mentioned at the beginning

Line 286-287: What are those female traits? Important to include here.

Line 290: the phenotypic engineering did not elongate the spines! It shortened them!! Am I missing something here?

Line 302-327: This explanation confirms my suspicion that males are changing the composition of their ejaculate when they are in this artificially enforced monogamy for a lifetime situation, and therefore the effect of the spine length is completely unknown! The increase in offspring production by females mated to Long spined males for life could be mediated by changes in ejaculate fluid as argued here.

Line 334-336. Exactly the problem here. The indirect benefit has to be through the descendants, and that was not shown in this experiment. You measured the same thing as direct and indirect benefits.

Line 368-371: While I agree wholeheartedly with the idea, this is not shown in this study.

Author's Response to Decision Letter for (RSPB-2021-0578.R0)

See Appendix A.

RSPB-2021-1068.R0

Review form: Reviewer 1

Recommendation

Accept as is

Scientific importance: Is the manuscript an original and important contribution to its field?
Excellent

General interest: Is the paper of sufficient general interest?

Good

Quality of the paper: Is the overall quality of the paper suitable?

Excellent

Is the length of the paper justified?

Yes

Should the paper be seen by a specialist statistical reviewer?

No

Do you have any concerns about statistical analyses in this paper? If so, please specify them explicitly in your report.

No

It is a condition of publication that authors make their supporting data, code and materials available - either as supplementary material or hosted in an external repository. Please rate, if applicable, the supporting data on the following criteria.

Is it accessible?

Yes

Is it clear?

Yes

Is it adequate?

Yes

Do you have any ethical concerns with this paper?

No

Comments to the Author

I thank the authors for responding to my comments. Happily, I feel the authors have addressed all of my comments in a satisfactory way. On second reading, I think this really is a great paper, which has become much more apparent after the different sources of data have been clarified. I am also much happier with the exclusion of the one outlier female after her behaviour was described in more detail.

I have just a few small suggestions:

1. You should add the species name somewhere in the abstract
2. Lines 189- 190: I think you should rephrase this to something like "both the original and additional experiments were run in 2012". As it is, the sentence is still a little ambiguous
3. Lines 375-377. Would it be correct to spell out this result even more plainly as "females mated to long-spine males lines have daughters that are larger and have higher fecundity"?

Review form: Reviewer 3

Recommendation

Accept as is

Scientific importance: Is the manuscript an original and important contribution to its field?
Excellent

General interest: Is the paper of sufficient general interest?
Good

Quality of the paper: Is the overall quality of the paper suitable?
Excellent

Is the length of the paper justified?
Yes

Should the paper be seen by a specialist statistical reviewer?
Yes

Do you have any concerns about statistical analyses in this paper? If so, please specify them explicitly in your report.
No

It is a condition of publication that authors make their supporting data, code and materials available - either as supplementary material or hosted in an external repository. Please rate, if applicable, the supporting data on the following criteria.

Is it accessible?
Yes

Is it clear?
Yes

Is it adequate?
Yes

Do you have any ethical concerns with this paper?
No

Comments to the Author

This paper examines the role of harmful male genital spines on female fitness in seed beetles, and presents significant results. I am happy the authors resurrected the data from this second set of experiments (following Hotzy et al 2012), as the results herein significantly improve our understanding the interplay of costs and benefits in sexual coevolution. It is exciting to see actual, measured indirect genetic benefits in females, especially when they are paired with direct costs. I believe this iteration of the manuscript has addressed the concerns raised in previous review. In particular I appreciate the efforts you have gone to in explaining the various experiments and how they differ from the Hotzy et al (2012) study. This has been a difficult paper to digest, but once digested highly rewarding. This is a significant contribution to our understanding of the interplay of competing selective forces in driving the evolution of mating systems. Thank you.

Decision letter (RSPB-2021-1068.R0)

01-Jun-2021

Dear Professor Arnqvist

I am pleased to inform you that your manuscript RSPB-2021-1068 entitled "Direct and indirect effects of male genital elaboration in female seed beetles" has been accepted for publication in Proceedings B.

The referee(s) have recommended publication, but also suggest some minor revisions to your manuscript. Therefore, I invite you to respond to the referee(s)' comments and revise your manuscript. Because the schedule for publication is very tight, it is a condition of publication that you submit the revised version of your manuscript within 7 days. If you do not think you will be able to meet this date please let us know.

In order to ensure effective and robust dissemination and appropriate credit to authors the dataset(s) used should be fully cited. To ensure archived data are available to readers, authors

should include a 'data accessibility' section immediately after the acknowledgements section. This should list the database and accession number for all data from the article that has been made publicly available, for instance:

Sincerely,

Professor Gary Carvalho

Associate Editor

Board Member

Comments to Author:

Many thanks to the authors for their careful revision and thoughtful response letter. The reviewers and I all agree that the revised manuscript is much clearer. Reviewer 1 has some very minor suggestions, but I am otherwise happy to recommend acceptance.

Reviewer(s)' Comments to Author:

Referee: 1

Comments to the Author(s).

I thank the authors for responding to my comments. Happily, I feel the authors have addressed all of my comments in a satisfactory way. On second reading, I think this really is a great paper, which has become much more apparent after the different sources of data have been clarified. I am also much happier with the exclusion of the one outlier female after her behaviour was described in more detail.

I have just a few small suggestions:

1. You should add the species name somewhere in the abstract
2. Lines 189- 190: I think you should rephrase this to something like "both the original and additional experiments were run in 2012". As it is, the sentence is still a little ambiguous
3. Lines 375-377. Would it be correct to spell out this result even more plainly as "females mated to long-spine males lines have daughters that are larger and have higher fecundity"?

Referee: 3

Comments to the Author(s).

This paper examines the role of harmful male genital spines on female fitness in seed beetles, and presents significant results. I am happy the authors resurrected the data from this second set of experiments (following Hotzy et al 2012), as the results herein significantly improve our understanding the interplay of costs and benefits in sexual coevolution. It is exciting to see actual, measured indirect genetic benefits in females, especially when they are paired with direct costs. I believe this iteration of the manuscript has addressed the concerns raised in previous review. In particular I appreciate the efforts you have gone to in explaining the various experiments and how they differ from the Hotzy et al (2012) study. This has been a difficult paper to digest, but once digested highly rewarding. This is a significant contribution to our understanding of the interplay of competing selective forces in driving the evolution of mating systems. Thank you.

Author's Response to Decision Letter for (RSPB-2021-1068.R0)

See Appendix B.

Decision letter (RSPB-2021-1068.R1)

09-Jun-2021

Dear Professor Arnqvist

I am pleased to inform you that your manuscript entitled "Direct and indirect effects of male genital elaboration in female seed beetles" has been accepted for publication in Proceedings B.

Your article has been estimated as being 8 pages long. Our Production Office will be able to confirm the exact length at proof stage.

Data Accessibility section

Open Access

Paper charges

Sincerely,
Editor, Proceedings B
<mailto:proceedingsb@royalsociety.org>

Appendix A

Authors' detailed responses to comments given (MS RSPB-2021-0578)

We were very happy to see that all three readers of our manuscript found our contribution of great interest and potentially suitable for publication in Proc B.

We are also very grateful indeed for the thorough assessment of our work and for the effort spent by the reviewers on our behalf. We have now prepared a thoroughly revised version which, as you will see, is very closely aligned with the suggestions provided by the reviewers of the previous version. We feel that we have been able to accommodate all suggestions given and that our contribution has improved significantly as a result. We hope that you will agree. We stress here that if you in any case feel that our response is unclear or that additional revisions are needed to further improve our contribution, we would of course be more than happy to discuss such issues and to further revise our manuscript should you consider this desirable or necessary.

Please find our responses to all comments given detailed below. Here, the actual comments are in italicized and in blue font and our responses are given in normal black font, to increase clarity and to facilitate your editorial work. All changes and revisions are highlighted using track-changes in MS Word in a dedicated PDF copy of our manuscript, to make our revisions maximally transparent.

Associate Editor

The two reviewers were somewhat mixed in their enthusiasm, though both raise concerns over how much of the data and analyses presented here are novel, and how much has been previously published. I was also somewhat confused by this, and without a clearer understanding, I think it is quite difficult to determine the novelty presented in this manuscript by the authors. Both reviewers also ask for a more thorough context of previous empirical findings. Reviewer 1 also raises concerns about the removal of two outlier females from the analysis. Reviewer 2 is far more critical, and raises key points about the limits that the experimental design place on the interpretation. These are, in my opinion, valid, especially the lack of distinction between female fecundity and female offspring production.

Thank you for this summary. Indeed, although both readers saw much merit in our work, as the AE noted, both also requested a clearer delineation of exactly how our data relates to the previous study and data of Hotzy et al [ref 35]. We had actually strived to be clear over this in our original version, but the reviewers' reports shows that we had simply not succeeded in being sufficiently clear. We sincerely apologize for this. Please find the revised version substantially improved in clarity in this regard. Hopefully, it should now be clear to all readers exactly what our data and experiments are and how they relate to previous work in this model system. Please see our detailed responses to the reviewers below. As you will see, all data involving the selection lines are entirely novel and, of course, previously unpublished. We agree with the AE in that both reviewers gave series valid and constructive comments and we have revised our MS in close agreement with these comments. Again, please see our responses below.

Referee: 1

*The manuscript reports a series of experiments examining the direct and indirect effects that male genital spines have on female fitness in the seed beetle *Callosobruchus maculatus*. The conflict between males and females that arises due to this harmful genital trait have been well-studied in this species, though results are mixed. By experimentally shortening male spines, the authors show for the first time that males with long spines significantly reduce female fecundity. However, artificial selection on male spine length finds the opposite result—reference females mating to males with long spines have higher fecundity, and selection for increased male spine length led to a correlated increase in female fecundity. These results suggest that females actually gain indirect benefits to mating with males with long spines, and that sexual coevolution in this species is driven by both sexual antagonism and good genes sexual selection.*

Overall, this is a very well-presented study describing an elegant series of experiments testing these ideas. Further, the results significantly improve our understanding of sexual coevolution in this model species, and will be relevant to the study of sexual conflict and sexual selection more generally. However, I have some major concerns with how the study is presented, and how it links to previous studies, which I feel the authors need to address.

We were very happy to see that the reviewer also found our results interesting and important for our understanding of sexual coevolution in our model system as well as for sexual selection more generally. Needless to say, we share this view. We have addressed all of the reviewer's comments in our revision.

Main comments

1. This study builds heavily from a study from this group from 2012 (Hotzy et al., 2012). This means that several of the approaches used in this study are not novel (though some of the results are). Specifically, the previous study used the same phenotypic engineering of male spine length (but found no significant difference between manipulated and control males), and compared the same experimental evolution lines (though female fitness was not recorded in the earlier study). This in and of itself is not a problem. However, for me this relationship was not made very transparent in the present paper, at least in the introduction (e.g. lines 105-111). I would go so far as to say that, for a reader not familiar with this previous study, this link would very much not be apparent from a casual read of the paper. So, to give due credit to this earlier work, and to put this study in its proper context, I think the links between these two studies needs to be much clearer in the introduction, and throughout.

We thank the reviewer for this general comment – as said above, we are most sympathetic to this request (see our comments to the AE above). We actually felt that this was clear in the original version, but on a second reading we now realize that these aspects were not at all sufficiently clear (in part, we note, reflecting an effort to restrict the length of the text). As a result of this comment, you will now find the text revised in many places in the MS (including but not restricted to former lines 105-111), with the explicit purpose of increasing

clarity. Hopefully, you will now find the methods maximally transparent. Several sections have been entirely rewritten.

Briefly, as the reviewer correctly notes, the selection lines used are the same selection lines previously used in distinct assays focused on male competitive fertilization success. During the publication of that work (Hotzy et al., 2012), we conducted two additional, novel and independent assays of female fitness during 2012 (!) before the lines were discontinued. However, the results of the new experiments were, due to a series of unfortunate reasons, left in the freezer and the file drawer for several years. These have now been processed. It is the results of these additional experiments (none of which have previously been analysed or published, of course) that we now present here. We agree with the reviewer that the fact that some time has passed since the collection of data cannot, of course, in and of itself be a problem.

2. Related to above, at several points that manuscript mentions using partly the same data from Hotzy et al., 2012 (lines 128-130, 240-242). However, it is not clear to me exactly which data were re-used. This needs to be made clearer here and in the introduction, along with a justification for why this way done.

As said above, we agree entirely. Please find these issues and analyses greatly clarified and motivated in our revised version.

3. More generally, I found the discussion of previous empirical studies in this species to be lacking. There is now a large literature focusing on sexual conflict in this species, and the results are complex. However, I feel some discussion of this complexity is useful for putting these results into context. For example, Cayetano & Bonduriansky (2015) explicitly test whether male genital spine length was condition dependent, but did not find an effect. This result is not mentioned when considering condition-dependence in the introduction (lines 88-89), or in the discussion (line 323). I would therefore like to see an expanded discussion of this and other relevant studies (incl. Hotzy et al., 2012) to better put these results into context. Including this context would not diminish the results of this study- indeed, the surprising result of this study (that both sexual conflict and good-genes effects may influence spine evolution) may partly explain why results of previous studies have been so mixed.

Point well taken. The original version reflected an effort to restrict the length of the text. We agree, however, and have thus somewhat expanded the discussion of these system-specific topics in our MS, for example in the positions highlighted by the reviewer. We note that we had already cited Cayetano & Bonduriansky (2015) in our original version, but we have now lifted this ref to the introduction as suggested.

4. I'm not entirely comfortable with the removal of the two outlier females in the phenotypic engineering experiment (lines 162-164), especially given that the result this pertains to is barely significant. Is $R > 2.5$ a common metric? Regardless, I would prefer you to present the results without removing these females.

Assessing and dealing with outliers can, depending on the data, be a critical part of sound data analysis. If, for example, a sick or otherwise aberrant female is included in a study of

the effects of treatment on a response, then the inclusion of this female can seriously bias the estimate of the effect of the treatment. In our original version, we used an absolute value of the standardized residual $R > 2.5$. This is a commonly used cutoff for medium sample sizes, and some 98.75% of all “sound” observations should fall within this delineation under the standard normal expectations. The probability of finding at least one “sound” female residing outside this delineation when sampling 48 females (as we did) is $P = 0.45$ [i.e., $1 - (0.9875^{48})$].

The reviewer is, however, “not entirely comfortable with this”. To meet this concern, we have dramatically increased the delineation of outliers to, in effect, an absolute value of the standardized residual $R > 5.0$. Here, the probability of finding at least one “sound” female residing outside this new delineation when sampling 48 females is $P = 0.00002$. We can thus be absolutely certain that any such female is aberrant. This definition leaves a single female (instead of two as before) defined as an outlier ($R = 5.6$). This female laid only one single egg after her second mating, which was well outside the range of all other females (range = 13 – 87; average = 42.2, SD = 15.6). We feel that including this clearly aberrant female would generate a biased analysis and we would not feel comfortable doing so. We hope that you can agree that this is reasonable.

Trimming data sets from outliers but failing to report this is a widely acknowledged problem in science. We feel that acknowledgement of the importance of dealing with outliers for sound data analysis and explicit definitions of outliers are both essential, as is maximal transparency and clarity in this regard. Please find this detailed in the revised results section. As you will see, defining only one instead of two females as outliers did in no case affect our ability to reject null hypotheses. In fact, the P value for the test of the focal hypotheses was reduced (from $P = 0.046$ to $P = 0.029$).

5. Several interesting results are presented in the supplementary material, but it is not obvious why. Does the manuscript as it stands overrun the Proc B space limits? If not, I think you should add these methods and results to the main text

We chose to place the details of these, admittedly interesting, experiments in the SM because (1) they are truly supplemental [i.e., they do not address how genital spines affect females] and (2) we wish not to exceed the Proc B page limit, as the reviewer suspected. We have nevertheless tried to accommodate this comment by including a brief account of the main results in the text of the revised version. We hope you feel that this is an acceptable compromise.

Other comments

1. Line 104: remove extra ‘of’

Ok – fixed.

2. Lines 181-183: why were two males used here?

Primarily to elevate the net cost of mating.

3. *Lines 169-171: When were the experiments using the sexual-selection lines done? Around 2012?*

Yes, 2012. This has now been specified.

4. *Line 213: please list the maximal value so readers do not have to access the reference*

Good point – done!

5. *Lines 238-239: why no figure for this significant effect?*

Well, this would be a plot of two means (the numbers and SE's are given at the top of the paragraph), which we honestly feel would add little but length to our contribution.

6. *Lines 290-292: other authors have suggested that negative effects on female fitness may be hard to detect because they are very small when the species is at a coevolutionary equilibrium. Do these results counter that argument?*

Good point – we have included a brief discussion on this in this paragraph.

Referee: 2

Comments to the Author(s)

RSPB 2021-0578

Direct and indirect effects of male genital elaboration in seed beetles

This paper examines the effect of male genital spine morphology on direct and indirect female fitness. The results are potentially very important because the effects of sexual conflict have been wrongly assumed to be only through direct fitness effects, while clearly there can be an impact on indirect fitness. The paper relies on spine ablation data from a previous published experiment adding body size as a novel aspect of the work, and from selection experiments for short and long spines.

We are very happy that the reviewer recognizes the importance of our results and the urgency to communicate these findings, as they modulate and supplement the interpretation of sexual antagonism in an important model system.

While the questions asked in this manuscript are indeed exciting, I had a hard time following the different experiments and their results. There is a lot of simplification here which seems necessary to make the paper more readable, but this comes at the expense of clarity of which crosses the authors are talking about in specific areas. I had to re-read the methods and result several times to try and figure out exactly what was done here.

We apologize for this confusion – we fully acknowledge that the original version was not written in a sufficiently clear manner (see comments above). Hopefully, you will find clarity significantly improved in the revised version.

In addition, I have a hard time with the presentation of both direct costs and direct benefits, in that the direct benefits results come from experiments where females were kept with the same male in monogamy for life, a situation that is highly unlikely in nature, and may have resulted in males modifying other behaviors or seminal fluids, that would mitigate the damage that is normally inflicted by the spines in a single mating, and the authors present good evidence for this in the discussion. That means that the result of increased in offspring production could simply be the result of male modification of ejaculate fluid and have nothing to do with spine length.

Thank you for sharing these concerns. We make the following reflections. First, in the experiment on direct costs of spines, females were not “kept with the same male in monogamy for life”: they were mated twice with two virgin males each and then kept in isolation for oviposition. This was necessary to provide sufficient control of potentially confounding variables in this experiment, and means that the results could not have been due to these sorts of complications. Second, in the experiments on the direct effects of the evolution of male spines, females were also not “kept with the same male in monogamy for life”. In the experiment involving base line females, each female was kept with two males for life. In the experiment involving selection lines females, five males and five females were kept together for life. The two experiments showed congruent results (i.e., females kept with L males produced more offspring – see MS). Third, although the particular complications that the reviewer delineates can thus not be an issue, we agree that the effects of evolutionary elaboration of the spines are very likely mediated by seminal fluid, as we suggest and discuss at some length in the discussion.

The section on indirect fitness is relies on female offspring production which is indistinguishable from fecundity as measured for direct benefits, with no justification of how these are interpreted as being different. While I fully understand the limitations of measuring the fitness of offspring produced by the experiments, that would have been the appropriate measure, and in that sense the results are oversold. The full results of all the statistical models should be included in supplementary materials.

We are struggling to interpret this comment, and suspect that it might reflect a misunderstanding of what we did. The section on indirect effects in females is based on the correlated response in females to selection in males across multiple generations. It is an entirely genetic effect: when we select for males with long/short spines over successive generations, we see the evolution of males with long/short spines but also the simultaneous correlated evolution of females with high/low offspring production. This is definitely a highly appropriate (some would argue the ultimate) measure of indirect genetic effects in females. To our knowledge, ours is the first study to achieve this for any male trait.

Line 105-106: How does this experiment overcome these challenges though? You are still only manipulating males so the confounding effects of female resistance and other male behaviors remain.

Phenotypic manipulation/engineering of sexual traits is a common strategy to isolate the effects of the manipulated trait in one sex, while keeping “all else constant”. To “only

manipulate males”, and to do so with only one trait, is in fact the very point of the experiment. The classic example is of course Malte Andersson’s 1982 study of widowbirds (cited 1026 times), where he prolonged and shortened male tail length and showed that female prefer to mate with males with long tails (rather than some trait that correlated with tail length). See e.g. Travis & Reznick (1998) [Experimental approaches to the study of evolution. In: Experimental ecology: issues and perspectives, 437-459] for a discussion of the basic principles and strengths of phenotypic manipulation experiments.

Lines 127-128. If the experiments are distinct, how can the data be the same? This is very confusing!

We fully agree that this was confusing and we hope these points have now been greatly clarified in the revised version (see our responses above).

Line 139-140. This study should stand on its own. Please provide brief description of those procedures using the reference for further details, not instead of an explanation.

Thank you for this comment. This section has now been rewritten to improve clarity and transparency.

Line 163. Please conduct the analyses with and without your outliers. Outliers are part of the normal biological variation and in an experiment with relatively small sample sizes such as this one, they may change the outcome of the statistics altogether.

This has now been revised, and data has been re-analysed. Please see our detailed response to a similar comment made by reviewer 1 above.

Lines 180-188. What is baseline female? A female from a non-selected line? And then what type of selected males? S, L both? There are 20 replicates per selection and then you selected one female to mate with males from each of these replicates? Why are there six lines? I thought there were 2 lines with 20 replicates each? Confusing! Please describe better.

Thanks for this comment. We have now edited this text to increase clarity.

Line 217. The questions that were meant to be addressed by this experiment are all important and relevant to the interpretation of the current results presented in this paper. This experiment should not be part of the supplementary materials, but should be included in full in the manuscript.

To reiterate our response to reviewer 1 above: “We chose to place the details of these, admittedly interesting, experiments in the SM because (1) they are truly supplemental [i.e., they do not address how genital spines affect females] and (2) we wish not to exceed the Proc B page limit, as the reviewer suspected. We have tried to accommodate this comment by including a summary of the main results in the text of the revised version. We hope you feel that this is an acceptable balance.”

Line 233. Name the covariates here. I believe they are male body size, female body size and number of eggs laid between matings?

Good point – done! Yes, that it correct.

Line 248: Average lifetime fitness production is 17% higher in L mated females than S mated females? How does that happen when the females produce 15% fewer eggs when mated with L males?

As we detail in the discussion, the former is a genetic response to the evolution of long spines in males, the latter a results of direct phenotypic manipulation of male spine length. Thus, the former must involve genetic effects which are correlated with male spine length.

Line 263: Females from L lines had a 6% higher production of offspring regardless of the phenotype of the male with whom they mated right?

Yes, this is averaged over all mates (i.e., are marginal means). This has now been clarified.

Line 269-270. How would this genetic correlation come to be? It could be that in lines where males have longer spines, female defenses such as kicking become more important and larger females are better at kicking...

Any of a number of scenarios could result in such a genetic covariance. The precise effect suggested here is unlikely, as the estimate of the genetic covariance is not based on the selection lines. This has now been clarified. But the general effect suggested (basically, a positive genetic covariance between male persistence and female resistance) is predicted from theory and could contribute to the sex-specific covariance seen. We would, however, prefer to not speculate on the many possible causes of this covariance in the interest of restricting the length of our contribution.

The “direct benefits” need further justification. How likely would it be in nature that a female would become fully monogamous and end up mating with a single male per life over multiple reproductive bouts? This is one of the confounding factors that the authors mentioned at the beginning

As detailed above, our experiments were not based on assaying females that “mate with a single male per life”. Please see our comments above.

Line 286-287: What are those female traits? Important to include here.

Unfortunately, we do not know what the cryptic female choice traits are. This is, as far as we know, not known in any system. This sentence is meant to be a general discussion and we have now edited the text to clarify this.

Line 290: the phenotypic engineering did not elongate the spines! It shortened them!! Am I missing something here?

Correct – thanks for pointing this out! This has now been rephrased.

Line 302-327: This explanation confirms my suspicion that males are changing the composition of their ejaculate when they are in this artificially enforced monogamy for a lifetime situation, and therefore the effect of the spine length is completely unknown! The increase in offspring production by females mated to Long spined males for life could be mediated by changes in ejaculate fluid as argued here.

Well, females were not kept under “enforced monogamy for a lifetime situation” (see our responses above) and the effects of spines per se is not “completely unknown” – we show here that the direct effect of spines in females is in fact negative. But, yes, the effects of evolution of long male spines in their mates (which is positive) very likely involves effects of male seminal fluid (as we suggest here).

Line 334-336. Exactly the problem here. The indirect benefit has to be through the descendants, and that was not shown in this experiment. You measured the same thing as direct and indirect benefits.

We are not sure what problem the reviewer is referring to here. The point and strength of our work, in this regard, is that we actually did measure indirect effects through the descendants. We suspect that the reviewer has somewhat misunderstood what we actually did, and have made efforts to clarify this in text of the revised version.

Line 368-371: While I agree wholeheartedly with the idea, this is not shown in this study.

Again, we suspect that this critical remark reflects at least a partial misunderstanding of what we achieved: we are measuring the indirect effect in females as a truly genetic effect (as a correlated response over many generations to sex-limited selection in males).

Appendix B

1

**Direct and indirect effects of male genital elaboration in female seed beetles**

Göran Arnqvist^{1*}, Karl Grieshop^{1,2}, Cosima Hotzy¹, Johanna Rönn¹, Michal Polak³ and Locke
Rowe^{2,4}

¹ Animal Ecology, Department of Ecology and Genetics, Evolutionary Biology Centre, Uppsala
University, Uppsala, Sweden.

² Department of Ecology and Evolutionary Biology, University of Toronto, Toronto, ON,
Canada.

³ Department of Biological Sciences, University of Cincinnati,
Cincinnati, OH 45221, USA.

⁴ Swedish Collegium for Advanced Study, Uppsala University, SE-752 38 Uppsala, Sweden

* Correspondence: goran.arnqvist@ebc.uu.se

Electronic supplementary material is available online at <https://doi.org/10.xxxx/xxxx/xxxx>.

**Abstract**

[revised manuscript text omitted]

Deleted: http://xxxxxxxx/xxxxxxxxxxxx/xx
 xxxxxxxx.

**Authors' contributions.** G.A., L.R. and K.G. conceived and designed the study, analysed and
 interpreted the data. G.A. and L.R. wrote the manuscript. K.G., C.H., M.P. and J.R. designed
 and performed experiments. All authors edited and approved the manuscript.

**Competing interests.** The authors declare that they have no competing
 interests.

**Funding.** This study was supported by grants from the European Research Council (GENCON
 AdG-294333), the Swedish Research Council (2014-4523; 2019-03611) and FORMAS (2018-

00705) to GA. LR was supported by the Natural Sciences and Engineering Research Council of
 Canada and the Swedish Collegium for Advanced Study. MP was supported by National
 Science Foundation (NSF) USA grant DEB-1654417. KG was supported by the Swedish
 Research Council (2018-06775).

**Acknowledgements.** We thank all members of the GENCON lab group, Uppsala University,
 for helpful discussions.

**References**

- 1. Kirkpatrick M, Ryan MJ. 1991 The evolution of mating preferences and the paradox of
 the lek. *Nature* **350**, 33-38.
- 2. Andersson MB. 1994 *Sexual selection*. Princeton, NJ: Princeton University Press.
- 3. Arnqvist G, Rowe L. 2005 *Sexual conflict*. Princeton, NJ: Princeton University Press.
- 4. Rice WR, Gavrilets S. (eds) 2014 *The genetics and biology of sexual conflict*. Cold Spring
 Harbor, NY: Cold Spring Harbor Laboratory Press.
- 5. Parker GA. 1979 Sexual selection and sexual conflict. In *Sexual selection and reproductive*
 *competition in insects* (eds Blum MS, Blum NB), pp. 123–166. New York: Academic Press.
- 6. Holland B, Rice WR. 1998 Chase-away sexual selection: antagonistic seduction versus
 resistance. *Evolution* **52**, 1–7.
- 7. Gavrilets S, Arnqvist G, Friberg U. 2001 The evolution of female mate choice by sexual
 conflict. *Proc. Roy. Soc. B.* **268**, 531-539.
- 8. Rowe L, Day T. 2006 Detecting sexual conflict and sexually antagonistic coevolution. *Phil.*
 *Trans. Roy. Soc. B.* **361**, 277-285.
- 9. Chenoweth SF, Appleton NC, Allen SL, Rundle HD. 2015 Genomic evidence that sexual
 selection impedes adaptation to a novel environment. *Curr. Biol.* **25**, 1860-1866.
- 10. Qvarnström A, Brommer JE, Gustafsson L. 2006 Testing the genetics underlying the co-
 evolution of mate choice and ornament in the wild. *Nature* **441**, 84–86.
- 11. Foerster K, Coulson T, Sheldon BC, Pemberton JM, Clutton-Brock TH, Kruuk LE. 2007
 Sexually antagonistic genetic variation for fitness in red deer. *Nature* **447**, 1107–1110.
- 12. Fricke C, Arnqvist G. 2007 Rapid adaptation to a novel host in a seed beetle
 (*Callosobruchus maculatus*): the role of sexual selection. *Evolution* **61**, 440–454.

- 13. Hollis B, Houle D. 2011 Populations with elevated mutation load do not benefit from the
operation of sexual selection. *J. Evol. Biol.* **24**, 1918–1926.
- 14. Arbuthnott D, Rundle, HD. 2012 Sexual selection is ineffectual or inhibits the purging of
deleterious mutations in *Drosophila melanogaster*. *Evolution* **66**, 2127–2137.
- 15. Long TAF, Agrawal A, Rowe L. 2012 The effect of sexual selection on offspring fitness
depends on the nature of genetic variation. *Curr. Biol.* **22**, 204-208.
- 16. Arnqvist G. 2006 Sensory exploitation and sexual conflict. *Phil. Trans. Roy. Soc. B* **361**,
375–386.
- 17. Cameron E, Day T, Rowe L. 2003 Sexual conflict and indirect benefits. *J. Evol. Biol.* **16**,
1055-1060.
- 18. Arnqvist G. 2014 Cryptic female choice. In *The evolution of insect mating systems* (eds
Shuker D, Simmons L), pp. 204-220. Oxford: Oxford University Press.
- 19. Kirkpatrick M, Barton NH. 1997 The strength of indirect selection on female mating
preferences. *Proc. Natl. Acad. Sci.* **94**, 1282-1286.
- 20. Andersson M. 1982 Sexual selection, natural selection and quality advertisement. *Biol. J.*
*Linn. Soc.* **17**, 375-393.
- 21. Nur N, Hasson O. 1984 Phenotypic plasticity and the handicap principle. *J. Theor. Biol.*
**110**, 275-297.
- 22. Andersson M. 1986 Evolution of condition-dependent sex ornaments and mating
preferences: sexual selection based on viability differences. *Evolution* **40**, 804-816.
- 23. Rowe L, Houle D. 1996 The lek paradox and the capture of genetic variance by condition
dependent traits. *Proc. Roy. Soc. B.* **263**, 1415–1421.
- 24. Crudgington HS, Siva-Jothy MT. 2000 Genital damage, kicking and early death. *Nature*
**407**, 855-6.
- 25. Hotzy C, Arnqvist G. 2009 Sperm competition favors harmful males in seed beetles. *Curr.*
*Biol.* **19**, 404-407.
- 26. Rönn J, Katvala M, Arnqvist G. 2007 Coevolution between harmful male genitalia and
female resistance in seed beetles. *Proc. Nat. Acad. Sci.* **104**, 10921-10925.
- 27. Dougherty LR, van Lieshout E, McNamara KB, Moschilla JA, Arnqvist G, Simmons LW.
2017 Sexual conflict and correlated evolution between male persistence and female
resistance traits in the seed beetle *Callosobruchus maculatus*. *Proc. Roy. Soc. B.* **284**,
20170132.

[revised manuscript text omitted]